# Simulations and Experiments on the Vibrational Characteristics of Cylindrical Resonators with First Three Harmonic Errors

**DOI:** 10.3390/mi13101679

**Published:** 2022-10-06

**Authors:** Chen Liang, Kaiyong Yang, Yao Pan, Yunfeng Tao, Jingyu Li, Shilong Jin, Hui Luo

**Affiliations:** College of Advanced Interdisciplinary Studies, National University of Defense Technology, Changsha 410073, China

**Keywords:** CRG, cylindrical resonator, first three harmonics, frequency split

## Abstract

A cylindrical resonator gyroscope is a kind of Coriolis gyroscope, which measures angular velocity or angle via processing of the standing wave. The symmetry of a cylindrical resonator is destroyed by different degrees of geometric nonuniformity and structural damage in the machining process. The uneven mass distribution caused by the asymmetry of the resonator can be expressed in the form of a Fourier series. The first three harmonics will reduce the anti-interference ability of the resonator to the external vibration, as well as increase the angular random walk and zero-bias drift of the gyroscope. In this paper, the frequency split of different modes caused by the first three harmonic errors and the displacement of the center of the cylindrical resonator bottom plate are obtained by simulation, and the relationship between them is explored. The experimental results on five fused silica cylindrical resonators are consistent with the simulation, confirming the linear relationship between the n = 1 frequency split and second harmonic error. A method for evaluating the first three harmonic errors of fused silica cylindrical resonators is provided.

## 1. Introduction

An inertial navigation system is an autonomous navigation system that does not rely on external information [1]. A gyroscope is the key device of inertial navigation system, which directly determines the cost and performance of an inertial navigation system. A Coriolis vibratory gyroscope measures the angular velocity or angle via the processing of the standing waves, the representatives of which includes the hemispherical resonator gyroscope (HRG), the cylindrical resonator gyroscope (CRG), and the tuning fork gyroscope [2]. Compared with mechanical rotor gyroscopes and optical gyroscopes, the cylindrical resonator gyroscope has several advantages. Firstly, it has no wear element, which guarantees a long life. In addition, due to the highly symmetric structure, the CRG has outstanding characteristics of a large operating temperature range and insensitivity to overload. Moreover, because of its low cost, small size, high precision, and good stability, CRG is very suitable for medium- to high-precision applications such as stabilization systems for various platforms, remotely operated vehicles, and automated underwater vehicles [3].

The cylindrical resonator is the core component of a cylindrical resonator gyroscope, whose quality has a crucial impact on the performance of the CRG. The performance of the resonator can be improved by optimizing the resonator structure, using appropriate materials, and improving the manufacturing technology [4,5,6,7]. Metal materials have been widely used in resonator manufacturing for their low cost and easy processing [8,9]. However, the internal damping of metals is usually too high to realize high-Q resonators. Fused silica material is an important material for high-performance resonator gyroscopes because of its low internal damping, small coefficient of thermal expansion, and high isotropy, despite it requiring complex processing technology and higher manufacturing costs compared to metal materials [5,9,10].

The actual resonator as-fabricated usually has defects of geometrical nonuniformity and structural damage. These defects can be equivalent to the nonuniform distribution of the resonator mass which is defined as unbalanced mass [11]. The unbalanced mass can be expressed in the form of a Fourier series [12]. The harmonic error is the error component of the unbalanced mass under different series in terms of the sine and cosine. The fourth harmonic error leads to the originally equal frequency of the n = 2 mode splitting into a different two. The first three harmonics lead to the uneven damping distribution and increase the sensitivity of the resonator to external vibration. The external random vibration increases the angular random walk of the gyroscope, and the external harmonic vibration increases the drift of the gyroscope, resulting in a significant decline in the accuracy and the vibration immunity of the gyroscope [13,14,15,16]. Therefore, reducing the first four harmonic errors is an important task in the manufacturing process of the CRG, which is of great significance to improve the accuracy and the environment immunity of the CRG.

Studies on the fourth harmonic are quite intensive; however, due to the complexity in both theoretical aspects and practical experimental aspects, studies on the first three harmonic errors are rare [17,18,19,20,21]. Hanhwa Corp disclosed a force-measuring device for mass unevenness measurement of a hemispherical resonator [22]. Lopatin et al. from Russia proposed a measurement method to detect the first three harmonic errors of the hemispherical resonator by measuring the vibration displacement of the hemisphere resonator shell [23]. Bodunov and others evaluated the first three harmonic errors by measuring the vibration of the external stem with piezoelectric sensors [24]. This study aims to reveal and verify the relationship between the harmonic errors and frequency split of the cylindrical resonator and the center displacement of the bottom surface of the resonator stem through simulation and experiments, and to propose a method for identifying the first three harmonic errors of the cylindrical resonator.

In this article, the vibrational characteristics of cylindrical resonators with the first three harmonic errors are studied through theoretical analysis, simulation, and experiment. Firstly, a mathematical model of a resonator with unbalanced mass is established. Then, we create a finite element model through Comsol to investigate the frequency split and the displacement of the stem center. The relationship between the frequency split in n = 1 mode and the center displacement of the bottom surface of the cylindrical resonator with the second harmonic error is obtained through analyzing the simulation results. Lastly, experiments on five fused silica cylindrical resonators are conducted. The experimental results are consistent with the simulation results. For the second harmonic error, it can be reflected by frequency split in n = 1 mode and can also be reflected by the vibration of the resonator stem in n = 2 mode.

## 2. Mathematical Model

The resonator is considered as a thin elastic ring that vibrates only in its own plane, and a ring model with unbalanced mass is established as shown in Figure 1. The axial length of the ring is *L*, the mean radius of the ring is *R*, and *φ_i_* is the central angle where the unbalanced mass *m_i_* is distributed.

The unbalanced mass has an impact on the natural frequency and mode position of the ring. The natural frequency of the imperfect ring with mass point can be expressed as follows [25]:(1)$$f_{n_{1},n_{2}}^{2}=f_{n_{0}}^{2}/\left[1+\sum_{i}\frac{m_{i}}{M_{0}}\pm \frac{\alpha_{n}^{2}-1}{M_{0}(\alpha_{n}^{2}+1)}\sqrt{{\left(\sum_{i}m_{i}\cos2n\varphi_{i}\right)}^{2}+{\left(\sum_{i}m_{i}\sin2n\varphi_{i}\right)}^{2}}\right],$$
where *f_n_*_1_, *f_n_*_2_ is the natural frequency of the *n*-th mode of the resonator with uneven mass, *f_n_*_0_ is the vibration frequency of the perfect resonator, *M*_0_ is the mass of the perfect resonator, and *α_n_* is the amplitude ratio of the radial and the tangential vibration.

As shown in Figure 2, the resonator has two vibration modes, forming two eigen-axes. The angle of these two eigenaxes is 45° in n = 2 mode. When the resonator vibrates along one of the eigenaxes, its eigen vibration frequency will reach the maximum and minimum. The difference between the two eigen frequencies is frequency split. The frequency split ∆*fn* of the *n*-th mode can be calculated using Equation (1) [14].
(2)$$\Delta f_{n}=f_{n1}-f_{n2}=\frac{f_{n_{0}}\left(\alpha_{n}^{2}-1\right)}{M_{0}\left(\alpha_{n}^{2}+1\right)}\sqrt{{\left(\sum_{i}m_{i}\cos2n\varphi_{i}\right)}^{2}+{\left(\sum_{i}m_{i}\sin2n\varphi_{i}\right)}^{2}}.$$

When analyzing the influence of resonator mass defect, we can introduce the unbalanced mass by the uneven distribution of the resonator density *ρ* in the central angle using a Fourier series [12]:(3)$$\rho \left(\varphi \right)=\rho_{0}+\sum_{i=1}^{\infty }\rho_{i}\cos\left(k\varphi -\varphi_{i}\right),$$
where *k* is the harmonic number of the resonator mass distribution, *ρ*_0_ is the density of the perfect resonator, and *ρ_i_* is the *i*-th harmonic error of the resonator density. With the increase in the number of unbalanced mass points, the mass defects can be expressed in the integral form. Equation (2) can be rewritten as the following form:(4)$$\Delta f_{n}=f_{n_{1}}-f_{n_{2}}=\frac{f_{n_{0}}R\left(\alpha_{n}^{2}-1\right)}{M_{0}\left(\alpha_{n}^{2}+1\right)}\sqrt{{\left(\int_{0}^{2\pi }\rho \left(\varphi \right)\cos2n\varphi d\varphi \right)}^{2}+{\left(\int_{0}^{2\pi }\rho \left(\varphi \right)\sin2n\varphi d\varphi \right)}^{2}}.$$

Substitute the harmonic error expression into the equation of the frequency split mentioned above; the relationship between frequency split in n = 1 mode and uneven density *ρ*_2_ can be expressed as
(5)$$\Delta f_{1}=\frac{f_{10}R\left(\alpha_{1}^{2}-1\right)\left|\rho_{2}\right|\pi }{M_{0}\left(\alpha_{1}^{2}+1\right)}.$$

The first three harmonics cause the vibration of the center of mass. The reaction force at the stem when an imperfect resonator vibrates at the n = 2 mode can be an be approximately written as the following expression through integration [26]:(6)$$\begin{array}{l} F_{x}=0.25a\omega^{2}\left[3V\rho_{1}\cos\left(2\theta -\varphi_{1}\right)+V\rho_{3}\cos\left(2\theta -3\varphi_{3}\right)\right]\sin\omega t\\ F_{y}=0.25a\omega^{2}\left[3V\rho_{1}\sin\left(2\theta -\varphi_{1}\right)-V\rho_{3}\sin\left(2\theta -3\varphi_{3}\right)\right]\sin\omega t\\ F_{z}=0.5a\omega^{2}V\rho_{2}\cos\left(2\theta -2\varphi_{2}\right)\sin\omega t \end{array}$$
where *a* and *ω* is the amplitude and angular frequency of the imperfect resonator, *V* is the volume of the resonator, and *φ_i_*_=1,2,3_ are the azimuth angles of the *i*-th mass defects.

According to Equations (5) and (6), it can be obtained that the transverse vibration is equivalent to applying excitation along the direction of the first and third harmonic errors. The longitudinal vibration results in the standing wave locating on the second harmonic. A larger second harmonic results in a more longitudinal displacement of the stem. The maximum value of longitudinal vibration displacement and the mode frequency split in n = 1 mode are both linearly related to the second harmonic. Therefore, when the frequency split in n = 1 mode is larger, the maximum value of the longitudinal displacement of the resonator stem is also larger.

## 3. Simulations

### 3.1. Establish the Resonator Model

Firstly, a perfect resonator without defects is established. The structure of the cylindrical fused silica resonator in this paper is shown in Figure 3. In order to make the simulation model as close to the experiments as possible, three constraints of contact at the outer surface of the cylindrical resonator stem are added. The material parameters are shown in Table 1.

A fixed constraint is applied on the outer cylindrical surface of the fixture. Next, the grid model of the cylindrical resonator is elaborately meshed. During the establishment of the finite element models, factors such as grid size, uniformity, and distribution density have great impact on the results of the simulation. The frequency split of the perfect resonator in n = 2 mode is reduced through proper grid division, such that the resonator can be considered as an ideal resonator without defects. As shown in Figure 4, the grid model includes 1,222,141 free tetrahedron elements. The computation time usually takes 5 min.

According to Equation (2), the uneven mass of the resonator can be expressed in the form of a Fourier series. Assuming the geometry of the resonator is perfect, the uneven mass can be equivalent to the nonuniformity of density. The distribution of imperfect density on resonator with the first three harmonics is shown in Figure 5. The nonuniform density distribution period is 2π, π, and 2π/3, respectively. 

### 3.2. Relationships between the First Three Harmonic Errors and the Frequency Splits

The influence of the first three harmonics on the frequency split of n = 1, 2, 3 modes is independently studied in this section. As shown in Figure 6, the influence of the first harmonic error on the frequency split of the n = 1 mode is much greater than that of the n = 2 and n = 3 modes. The frequency split of the n = 1 mode is approximately proportional to the square of the first harmonic error, and its relationship is obtained by parameter estimation using the least square method. The frequency splits of the n = 2 and the n = 3 mode increase slightly with the growth the of the first harmonic error. The frequency split of the n = 1, 2, 3 modes are 20.72 mHz, 14.89 mHz, and 6.30 mHz, respectively, when the first harmonic error is 10 kg/m^3^. 

The influence of the second harmonic error on the frequency split is shown in Figure 7. With the increase in the second harmonic error, the frequency split of the n = 1 mode increases rapidly, and the frequency splits in the n = 2 and n = 3 modes are only slightly changed. The frequency split of the n = 1 mode is approximately linear with the second harmonic error. The frequency split of the n = 1 mode is 12.45 mHz when the second harmonic error is 20 kg/m^3^. From the fitting results, the frequency split of the n = 1 mode is proportional to the second harmonic, which is consistent with the theoretical formula. In addition, although the first harmonic can also cause an increase in the n = 1 frequency split, its contribution is far less than the second harmonic. It is still in line with the theoretical model.

The variation of the frequency split of the n = 1, 2, 3 modes with varied amount of the third harmonic error is shown in Figure 8. The influence of the third harmonic error on the frequency split of the n = 3 mode is greater than that on the n = 1 and the n = 2 modes. In general, the third harmonic error has little influence on the frequency split of the n = 1, 2, 3 modes. The variation of the frequency split is 2.2 mHz when the coefficient of the third harmonic error increase from 0 kg/m^3^ to 20 kg/m^3^

### 3.3. Relationships between the Displacement of the Center of the Bottom Plate and the First Three Harmonic Errors

The influence of the first three harmonic errors on the displacement of the center of the cylindrical resonator bottom plate is studied in this section. Simulations on the imperfect resonator have been conducted to investigate the characteristics of the vibration of the bottom plate center. The resonator is excited with the same magnitude of force at the same angle. The excitation force is determined by actual experimental experience. The resonator vibrates at the natural frequency of the n = 2 mode. The displacement of the bottom plate center is extracted by the analysis of the frequency domain.

As shown in Figure 9, the relationship between the harmonic errors and the displacement of the bottom center is obtained. With the increase in the first and third harmonic errors, the displacement of the bottom center also increases. The total displacement of the center point of bottom plate and displacement components of the XOY plane increase more obviously, while the displacement component along the Z direction is almost unchanged. 

With the increase in the second harmonic error from 0 to 9 kg/m^3^, the displacement components of the XOY plane remain unchanged. On the other hand, the displacement component in the Z direction and the total displacement of the bottom plate center increase significantly. Moreover, when the second harmonic error continues to increase to 20 kg/m^3^, the displacement of concern decreases. According to the theoretical model, the first and third harmonic errors lead to the transverse vibration of the bottom plate, and the second harmonic error leads to the longitudinal vibration of the bottom plate. The results of the simulation are consistent with theory.

However, in practice, the measurement system, manufacturing imperfections, *Q* factor difference among different resonators, etc. will cause variation in absolute displacement of the resonator and, consequently, the variation in the displacement of the center of the resonator bottom. Therefore, it is necessary to remove the impact of the total displacement caused. The maximum displacement of the resonator under the n = 2 mode is about 500 nm, which is reasonably small compared with the size of the resonator. Therefore, the vibration of the cylindrical resonator can be regarded as a linear system, allowing us to calculate the relative displacement of the bottom center of the cylindrical resonator. the coefficient of harmonic displacement *η* is defined as
(7)$$\eta =\frac{D_{c}}{D_{{}^{max}}},$$
where *D_c_* is the displacement of the center of the bottom surface, and *D_max_* is the maximum displacement of the bottom surface.

As shown in Figure 10, *η* increases as the first three harmonic errors grow. The coefficient *η* in the XOY plane increased linearly with first and three harmonic errors, and the coefficient *η* along the Z direction increased linearly with the second harmonic error. The coefficient of harmonic vibration *η* represents the relevant displacement under different first three harmonic errors. Therefore, it can be used to weigh the degree the harmonic errors for resonators with different structures under different measurement conditions. 

## 4. Experiments

In this section, experiments are conducted on five fused silica cylindrical resonators machined with the same processing parameters. These resonators were chemically etched in NH_4_F_2_ solution. This operation can effectively remove the subsurface damage layer such as micro-cracks and scratches generated during the processing of resonators. The experimental set up is shown in Figure 11. The acoustic source excites the cylindrical resonator to the n = 2 wineglass mode, and the laser Doppler vibrometer (PSV-500, Polytec, Waldbronn, Germany) acquires the vibration information, which is then processed by the Polytec software. The fixture is used to fix these experimental resonators. These resonators are fixed with the same clamping condition. The position of the laser Doppler vibrometer is adjusted so that the laser emitted can be shot toward the center of the resonator bottom plate along the normal direction of the bottom surface of the cylindrical resonator. This operation is to ensure the displacement of the measurement of the center of the bottom in the Z direction of the bottom.

Firstly, the frequency split of the cylindrical resonator in n = 1 mode and its natural frequency in working mode are obtained using a frequency sweeping operation. Similarly, because the displacement of the cylindrical resonator is reasonable small in the actual experiment, we can regard it as a linear system. Therefore, the resonator is excited with the natural frequency of the resonator. The vibration rates *v_c_* and *v_max_* are obtained during the excitation. Similarly, the coefficient of harmonic vibration can be defined as
(8)$$\gamma =\frac{v_{c}}{v_{max}},$$
where *v_c_* is the rate at the center of the resonator, and *v_max_* is the rate at the maximum rate point on the bottom of the resonator.

There are 691 points set on the surface of the bottom plate. The result of the points scanning of one resonator is shown in Figure 12. The two points marked in the figure are the center point of the resonator and the point of the maximum rate.

The relationship between the frequency split of cylindrical resonator in n = 1 mode and the relative velocity of its bottom center is obtained through experiments. Table 2 shows the experimental data of five resonators. The unbalance mass distribution of resonators is different, resulting in different vibration states of resonators. The vibration frequency and bottom displacement of resonators may be significantly different. It can be seen from the experimental results that the coefficient of harmonic vibration and the frequency split of the resonator under n = 1 mode still conform to the linear law.

As Figure 13 shows, the second harmonic error has a linear relationship with the frequency split of the resonator in n = 1 mode. It is consistent with the simulation results.

## 5. Discussion

In summary, the vibration characteristics of the cylindrical resonator with the first three harmonic errors were studied in this paper. Through the finite element analysis, with the increase in harmonic errors, the frequency split of the cylindrical resonator in n = 1, 2, and 3 mode also increased on different levels. Among them, although the first harmonic error had a quadratic function relationship with the frequency split of the resonator in n = 1 mode; its effect was two orders of magnitude smaller than that of the second harmonic error. It was shown that the second harmonic error had a linear relationship with the frequency split of the resonator in n = 1 mode, while the third harmonic error had almost no impact. Therefore, it is proposed that the frequency split of the n = 1 mode can be regarded as a direct character to assess the second harmonic error of the cylindrical resonator.

On the other hand, with the increase in the first three harmonics, the displacement of the bottom center in the n = 2 mode increased. Simulation results showed that the displacement in the XOY plane was mainly caused by the first and the third harmonic errors, while the displacement along the Z direction was mainly caused by the second harmonic error. To eliminate the influence of the excitation conditions, different frequency and damping characteristics, etc., a relative displacement ratio *η* was introduced. It was shown that the second harmonic error had a linear relationship with *η*. In addition, the simulation showed a linear relationship between the frequency split of the resonator in the n = 1 mode and the relative displacement of its bottom center in the n = 2 mode.

In this paper, experimental results of five different cylindrical resonators were reported and found to be in line with the finite element simulation results. The results showed that the second harmonic error can be indirectly characterized by measuring the bottom center displacement of the cylindrical resonator, which makes it possible to quickly measure the first three harmonic errors of the cylindrical resonator.

## Figures and Tables

**Figure 1 micromachines-13-01679-f001:**
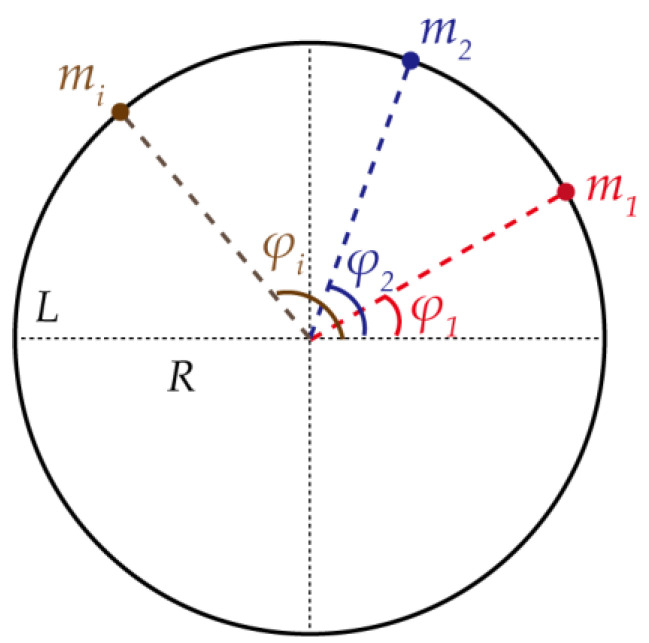
The ring with unbalanced mass.

**Figure 2 micromachines-13-01679-f002:**
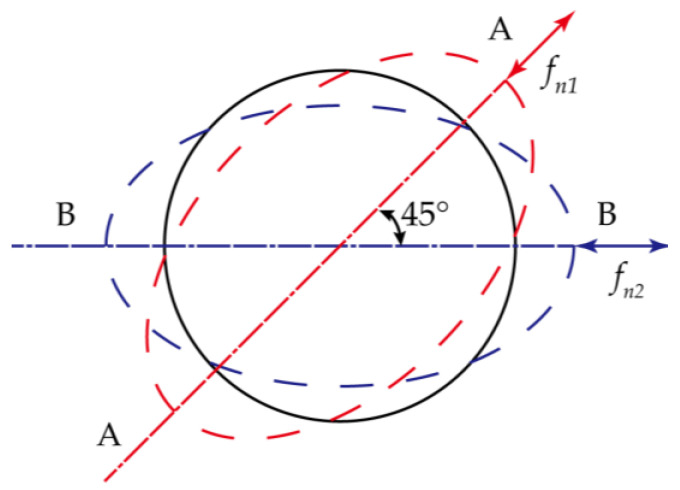
Schematic diagram of resonator vibration. AA, BB are the two eigenaxes. *fn*_1_, *fn*_2_ are the natural frequency of the resonator.

**Figure 3 micromachines-13-01679-f003:**
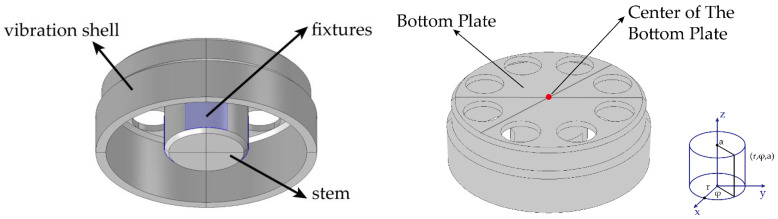
Structural diagram of cylindrical resonator with fixtures and the cylindrical coordinate system.

**Figure 4 micromachines-13-01679-f004:**
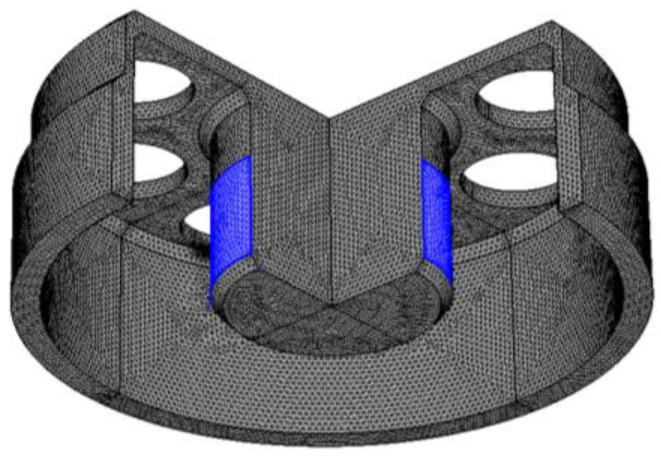
Grid model of the cylindrical resonator.

**Figure 5 micromachines-13-01679-f005:**
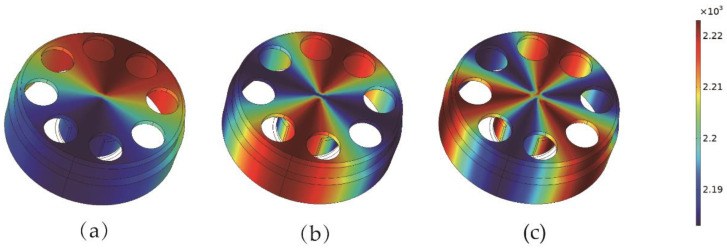
The distribution of the first three nonuniform densities: (**a**) the first harmonic error; (**b**) the second harmonic error; (**c**) the third harmonic error.

**Figure 6 micromachines-13-01679-f006:**
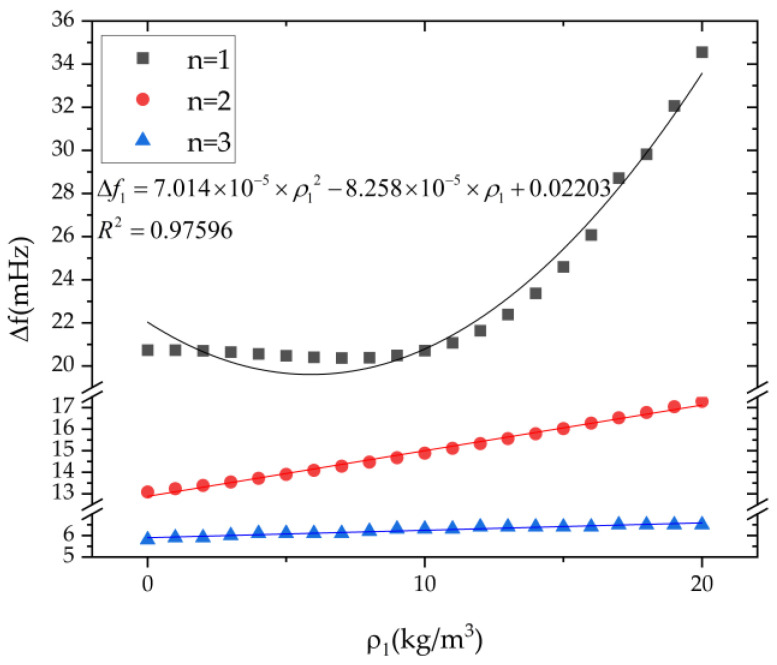
The influence of the first harmonic *ρ*_1_ on the frequency split of n = 1, 2, 3 modes.

**Figure 7 micromachines-13-01679-f007:**
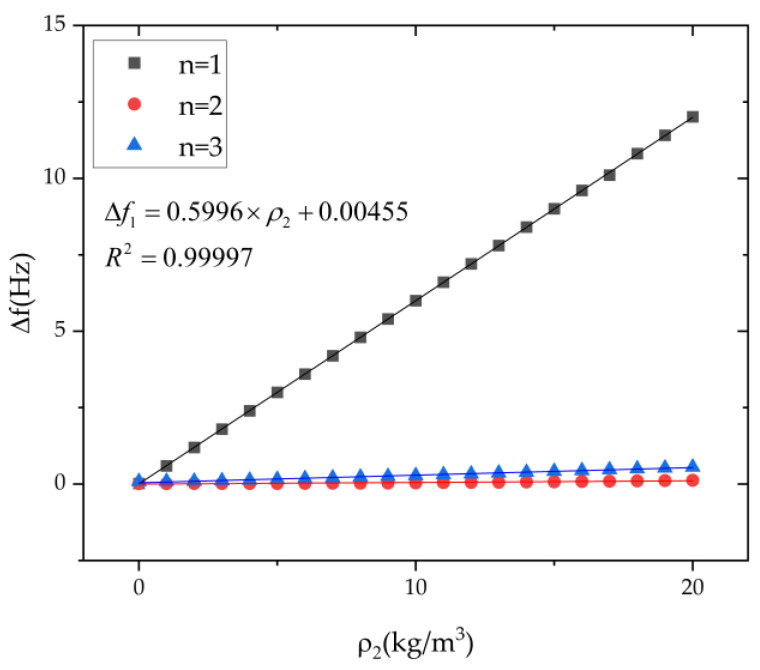
The influence of the second harmonic *ρ*_2_ on the frequency split of n = 1, 2, 3 modes.

**Figure 8 micromachines-13-01679-f008:**
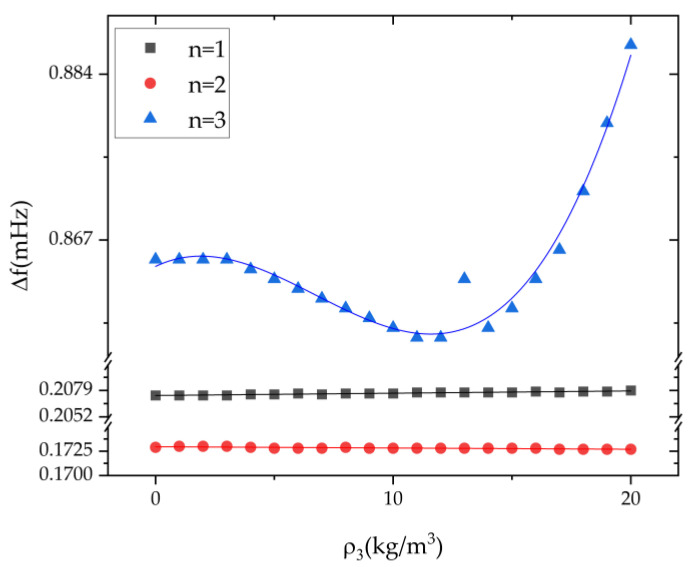
The influence of the third harmonic *ρ_3_* on the frequency split of n = 1, 2, 3 modes.

**Figure 9 micromachines-13-01679-f009:**
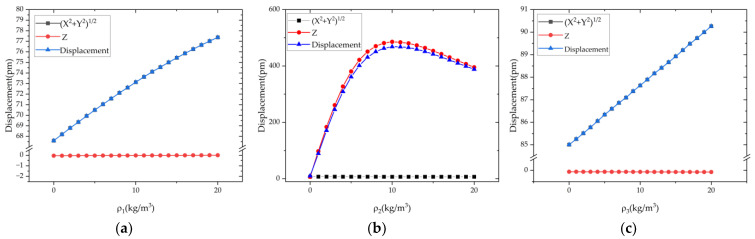
The influence of the first three harmonics *ρ*_1,2,3_ on the displacement of the center of the cylindrical resonator bottom plate: (**a**) *ρ*_1_; (**b**) *ρ*_2_; (**c**) *ρ*_3_.

**Figure 10 micromachines-13-01679-f010:**
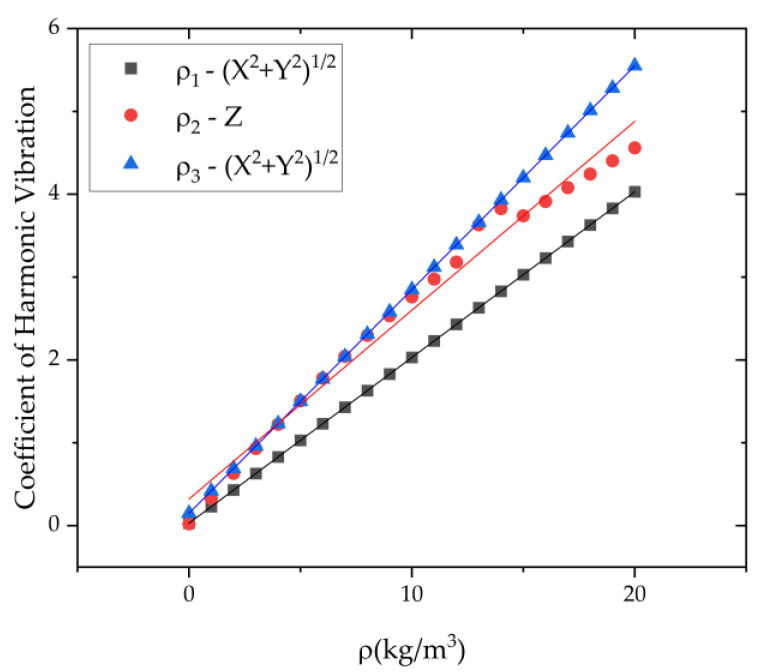
The coefficient of harmonic vibration under the first three harmonic errors *ρ*_1,2,3_.

**Figure 11 micromachines-13-01679-f011:**
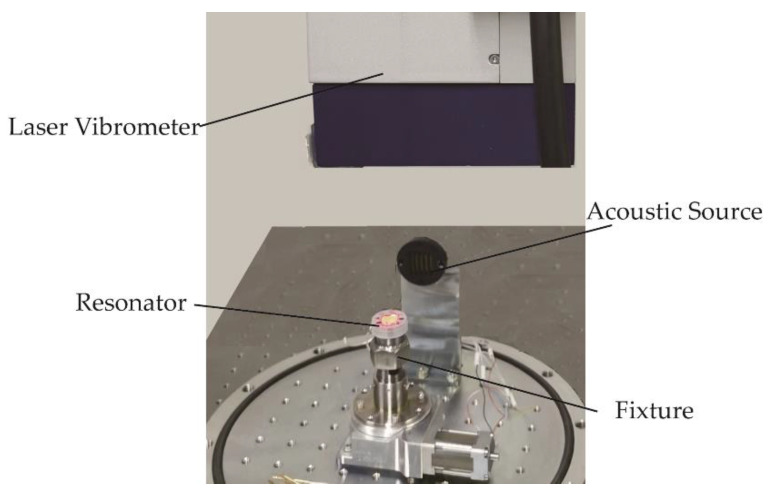
The experimental setup.

**Figure 12 micromachines-13-01679-f012:**
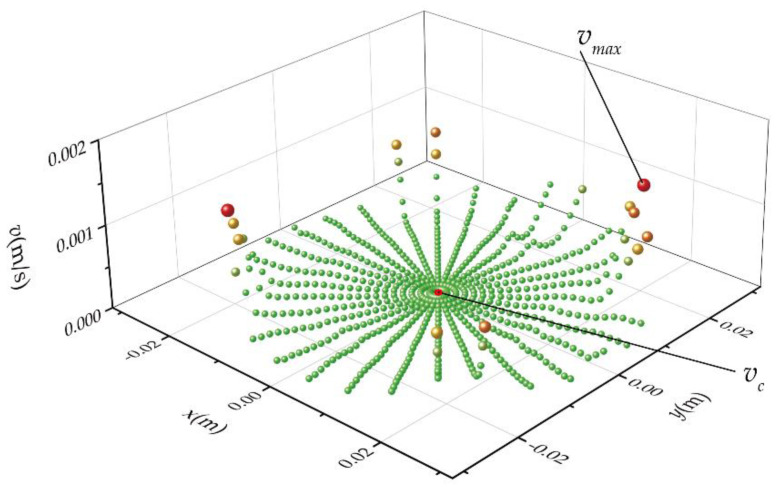
Scanning results diagram.

**Figure 13 micromachines-13-01679-f013:**
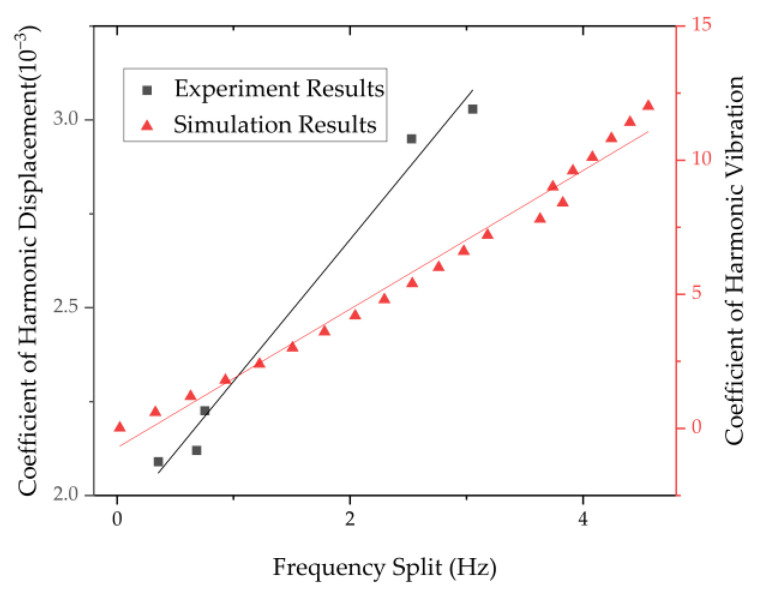
The relationship between the frequency split of cylindrical resonator in n = 1 mode and the coefficient of harmonic vibration.

**Table 1 micromachines-13-01679-t001:** Materials of the simulation model.

	Young’s Modulus (GPa)	Poisson’s Ratio	Density (kg/m^3^)
Resonator	71.7	0.17	2203
Fixture	90	0.32	8500

**Table 2 micromachines-13-01679-t002:** Vibration frequency, frequency split under n = 1 mode, and coefficient of harmonic vibration of experimental resonators.

Resonator Number	Vibration Frequency (Hz)	Frequency Under n = 1 Mode (Hz)		Frequency Split Under n = 1 (Hz)	*v_max_* (μm/s)	*v_c_* (μm/s)	Γ (10^−3^)
1	8078.748	3102.380	3104.907	2.527	968.2	2.858	2.95
2	8119.080	3129.590	3132.642	3.052	1167	3.535	3.029
3	8104.285	3071.240	3071.923	0.683	1278	2.710	2.12
4	8095.251	3110.010	3110.754	0.753	1176	2.618	2.226
5	8166.211	3397.705	3398.059	0.354	1156.9	2.418	2.09

## Data Availability

Not applicable.

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
