# Peer review of "Simulations and Experiments on the Vibrational Characteristics of Cylindrical Resonators with First Three Harmonic Errors"

_micromachines, 2022, doi:10.3390/mi13101679_

Round 1
Reviewer 1 Report
In this work, the author studied the first three harmonic errors of cylindrical resonators. I think this work can be published after following modification:
1) line 48-52, please give a clearer introduction of the idea of 'harmonic error'. Also, it will be better to add an brief explanation of why the first 4 orders are important.
2) line 227-228, the author referred the tested resonators are 'chemically treated', but no more information about it. Looks like there is a missing paragraph. Please give more detailed sample condition and testing setting.
3) the reference: please double check your references. The 16, 18, 19, 21 missed the journal name.
Author Response
Response to Reviewer 1 Comments
Point 1: line 48-52, please give a clearer introduction of the idea of 'harmonic error'. Also, it will be better to add an brief explanation of why the first 4 orders are important.
Response 1: Thank you for your valuable suggestion. We have added the following sentence to describe the definition of harmonic error more clearly, and state the importance of the reduction of first four harmonic errors.
Page 2, Line 49:
These defects can be equivalent to the nonuniform distribution of the resonator mass which is defined as unbalanced mass. The unbalanced mass can be expressed in the form of Fourier series. The harmonic error is the error component of the unbalanced mass under different series in terms of the sine and cosine.
Point 2: line 227-228, the author referred the tested resonators are 'chemically treated', but no more information about it. Looks like there is a missing paragraph. Please give more detailed sample condition and testing setting.
Response 2: Thank you for your valuable suggestion. We have added the following sentence to describe the chemical treatment process, and give more settings about the pretreatment of the resonator for experiment.
Page 8, Line 240:
In this part, experiments are conducted on five fused silica cylindrical resonators which have been machined with the same processing parameters. And these resonators have been chemically etched in NH4F2 solution. This operation can effectively remove the sub -surface damage layer such as micro -cracks and scratches generated during the processing of resonators.
Point 3: the reference: please double check your references. The 16, 18, 19, 21 missed the journal name..
Response 3: We are sorry for this mistake. We have checked and corrected the references.
Page 11, Line 332:
16. Rourke, A.K.; Mcwilliam, S.; Fox, C.H. Frequency trimming of a vibrating ring-based multi-axis rate sensor. Journal of Sound and Vibration 2005, 280, 495-530.
18. Hu, Z.; Gallacher, B.J.; Burdess, J.S.; Bowles, S.; Grigg, H.T.D. A systematic approach for precision electrostatic mode tuning of a MEMS gyroscope. Journal of Micromechanics Microengineering 2014, 24, 125003.
19. Pan, Y.; Tao, Y.; Zeng, L.; Tang, X.; Yang, K.; Luo, H. Investigation on the Optimal Fixation Condition of Cylindrical Resonators. 28th Saint Petersburg International Conference on Integrated Navigation Systems 2021, 1-3.
21. Wang, Y.; Pan, Y.; Qu, T.; Jia, Y.; Yang, K.; Luo, H. Decreasing Frequency Splits of Hemispherical Resonators by Chemical Etching. Sensors 2018, 18.

Reviewer 2 Report
Authors present a good paper on Simulations and experiments on the vibrational characteristics of cylindrical resonators with first three harmonic errors. State of the art is well presented. Simulation of frequency split of different modes caused by the first three harmonic errors and the displacement of the center of the cylindrical resonator bottom plate is proposed and well correlate experimental results given in the paper.
Some remarks to improve the paper
1/ Details of the EF model should be given, number of nodes, computing duration …
2/ Problems in l175, mode 3 instead of mode 2 and l176 influence
3/ In Fig. 13, Five experimental dots are shown. The big difference between three and two sets should be more explain. Table could give their characteristics.
Author Response
Response to Reviewer 2 Comments
Point 1: Details of the EF model should be given, number of nodes, computing duration.
Response 1: Thank you for your valuable suggestion. We have added the following sentence to describe the details of the resonator model
Page 5, Line 141:
During the establishment of the finite element models, factors such as grid size, uniformity, and distribution density have great impact on the results of the simulation. The frequency split of the perfect resonator in n=2 mode is reduced through proper grid division, so that the resonator can be considered as an ideal resonator without defects. As shown in Figure 4, the grid model includes 1,222,141 free tetrahedron elements. The computation time usually cause five minutes.
Point 2: Problems in l175, mode 3 instead of mode 2 and l176 influence.
Response 2: We are sorry for this mistake. We have corrected this mistake.
Page 6, Line 186:
The influence of the third harmonic error on the frequency split of the n=3 mode is greater than that on the n=1 and the n=2 modes.
Point 3: In Fig. 13, Five experimental dots are shown. The big difference between three and two sets should be more explain. Table could give their characteristics.
Response 3: Thank you for your valuable suggestion. The data in Figure 13 are the results of the actual test. The harmonic error values of the resonators used in the experiment differ greatly, which caused the differences of the experimental result. We have added the following sentences and table to explain the differences in experimental results better.
Page 10, Line 266:
Table 2 shows the experimental data of five resonators. The unbalance mass distribution of resonators is different, resulting in different vibration states of resonators. The vibration frequency and bottom displacement of resonators may be significantly different. It can be seen from the experimental results that the coefficient of harmonic vibration and the frequency split of the resonator under n=1 mode still conform to the linear law.
Table 2. Vibration frequency, frequency split under n=1 mode and coefficient of harmonic vibration of experimental resonators.
Resonator Number |
Vibration Frequency (Hz) |
Frequency Under n=1 Mode (Hz) |
Frequency Split Under n=1 (Hz) |
vmax(μm/s) |
vc(μm/s) |
γ(10-3) |
|
1 |
8078.748 |
3102.380 |
3104.907 |
2.527 |
968.2 |
2.858 |
2.95 |
2 |
8119.080 |
3129.590 |
3132.642 |
3.052 |
1167 |
3.535 |
3.029 |
3 |
8104.285 |
3071.240 |
3071.923 |
0.683 |
1278 |
2.710 |
2.12 |
4 |
8095.251 |
3110.010 |
3110.754 |
0.753 |
1176 |
2.618 |
2.226 |
5 |
8166.211 |
3397.705 |
3398.059 |
0.354 |
1156.9 |
2.418 |
2.09 |
